# High Prognostic Value of ^68^Ga-PSMA PET/CT in Renal Cell Carcinoma and Association with PSMA Expression Assessed by Immunohistochemistry

**DOI:** 10.3390/diagnostics13193082

**Published:** 2023-09-28

**Authors:** Donatello Gasparro, Maura Scarlattei, Enrico Maria Silini, Silvia Migliari, Giorgio Baldari, Veronica Cervati, Tiziano Graziani, Nicoletta Campanini, Umberto Maestroni, Livia Ruffini

**Affiliations:** 1Oncology Division, Azienda Ospedaliero-Universitaria di Parma, 43126 Parma, Italy; dgasparro@ao.pr.it; 2Nuclear Medicine Division, Azienda Ospedaliero-Universitaria di Parma, 43126 Parma, Italy; mscarlattei@ao.pr.it (M.S.); gbaldari@ao.pr.it (G.B.); vcervati@ao.pr.it (V.C.); tgraziani@ao.pr.it (T.G.); lruffini@ao.pr.it (L.R.); 3Pathology Division, Azienda Ospedaliero-Universitaria di Parma, 43126 Parma, Italy; enricomaria.silini@unipr.it (E.M.S.); nicoletta.campanini@unipr.it (N.C.); 4Department of Medicine and Surgery, University of Parma, 43121 Parma, Italy; 5Urology Division, Azienda Ospedaliero-Universitaria di Parma, 43126 Parma, Italy; umaestroni@ao.pr.it

**Keywords:** renal-cell carcinoma (RCC), PSMA expression, PET-CT

## Abstract

In oligo-metastatic renal cell carcinoma (RCC), neither computed tomography (CT) nor bone scan is sensitive enough to detect small tumor deposits hampering early treatment and potential cure. Prostate-specific membrane antigen (PSMA) is a transmembrane glycoprotein expressed in the neo-vasculature of numerous malignant neoplasms, including RCC, that can be targeted by positron emission tomography (PET) using PSMA-targeting radioligands. Our aim was to investigate whether PSMA-expression patterns of renal cancer in the primary tumor or metastatic lesions on immunohistochemistry (IHC) are associated with PET/CT findings using [^68^Ga]-PSMA-HBED-CC (PSMA-PET/CT). We then analyzed the predictive and prognostic role of the PSMA-PET/CT signal. In this retrospective single-center study we included patients with renal cancer submitted to PSMA-PET/CT for staging or restaging, with tumor specimens available for PSMA-IHC. Clinical information (age, tumor type, and grade) and IHC results from the primary tumor or metastases were collected. The intensity of PSMA expression at IHC was scored into four categories: 0: none; 1: weak; 2: moderate; 3: strong. PSMA expression was also graded according to the proportion of vessels involved (PSMA%) into four categories: 0: none; 1: 1–25%; 2: 25–50%; 3: >50%. The intensity of PSMA expression and PSMA% were combined in a three-grade score: 0–2 absent or mildly positive, 3–4 moderately positive, and 5–6 strongly positive. PSMA scores were used for correlation with PSMA-PET/CT results. Results: IHC and PET scans were available for the analysis in 26 patients (22 ccRCC, 2 papillary RCC, 1 chromophobe, 1 “not otherwise specified” RCC)**.** PSMA-PET/CT was positive in 17 (65%) and negative in 9 patients (35%). The mean and median SUVmax in the target lesion were 34.1 and 24.9, respectively. Reporter agreement was very high for both distant metastasis location and local recurrence (kappa 1, 100%). PSMA-PET detected more lesions than conventional imaging and revealed unknown metastases in 4 patients. Bone involvement, extension, and lesion number were greater than in the CT scan (median lesion number on PET/CT 3.5). The IHC PSMA score was concordant in primary tumors and metastases. All positive PSMA-PET/CT results (15/22 ccRCC, 1 papillary cancer type II, and 1 chromofobe type) were revealed in tumors with strong or moderate PSMA combined scores (3–4 and 5–6). In ccRCC tissue samples, PSMA expression was strong to moderate in 20/22 cases. The SUVmax values correlated to the intensity of PSMA expression which were assessed using IHC (*p* = 0.01), especially in the ccRCC subgroup (*p* = 0.009). Median survival was significantly higher in patients with negative PSMA-PET/CT (48 months) compared to patients with a positive scan (24 months, *p*= 0.001). SUVmax ≥ 7.4 provides discrimination of patients with a poor prognosis. Results of PSMA-PET/CT changed treatment planning. Conclusions: in renal cancer, positive PSMA-PET/CT is strongly correlated to the intensity of PSMA expression on immunohistochemistry in both ccRCC and chromophobe cancer. PSMA-PET/CT signal predicts a poor prognosis confirming its potential as an aggressiveness biomarker and providing paramount additional information influencing patient management.

## 1. Introduction

Renal cell carcinoma (RCC) is the most common tumor of the kidney, and its incidence has risen steadily, accounting for 4% of all malignancies [1]. Clear-cell (cc) RCC accounts for 70–80% of RCC subtypes. Less frequent subtypes include papillary (10–15%) and chromophobe RCCs (5%) [2].

Most patients have localized disease at diagnosis, and they undergo surgical treatment with curative intent. However, one-fourth of them will develop the metastatic disease with a 5-year survival rate below 10% [3,4], and 35–47% of patients with locally advanced (T2–T4) renal cell carcinoma (RCC) recur after surgery and develop metastasis.

Furthermore, up to one-third of patients with newly diagnosed RCC have metastatic diseases at presentation [5].

Hence the need for a global and accurate staging is emerging especially in younger individuals that have higher rates of multiple concomitant metastases [6,7] and in aggressive tumor types, even more so considering the results of the RECURR multicentre European registry which assessed the effectiveness of local treatment of recurrence in RCC [8].

Contrast-enhanced computed tomography (CT) and magnetic resonance imaging (MRI) remain the mainstay for RCC staging with almost equally high accuracy. For CT, median sensitivity and specificity were reported to be 88% and 75%; for MRI they were 87.5% and 89% [4,5,6,7,8].

In patients previously submitted to surgery or ablation, imaging aims at early detection of loco-regional recurrence or distant metastasis [9]. Early diagnosis of recurrence provides more opportunities for effective treatment, especially for oligometastatic patients [10]. Indeed, 30% of these may remain disease-free for 5 years after metastasectomy [11].

In the case of advanced disease, imaging may assess disease burden, response to systemic therapy, and disease course [12,13].

Moreover, innovative approaches such as active surveillance postponing systemic therapy or a combination of treatments determining a greater clinical benefit for patients when compared to a single drug [14,15,16,17,18,19,20] may increase the need for more accurate and earlier identification of candidates [21].

New therapeutic options for patients with recurrent or metastatic disease need to identify predictive biomarkers of efficacy or resistance.

Molecular imaging using targeted radioligands represent a powerful non-invasive tool for the functional characterization of tumors ‘in vivo’ with positron emission tomography (PET) [22].

The most used PET technique to assess cancer burden is metabolic imaging with [^18^F]Fluorodeoxyglucose (FDG) and several studies suggested that FDG-PET is valuable for detecting recurrent or metastatic foci of RCC. A systematic analysis of 15 eligible diagnostic trials involving 1168 patients showed that FDG-PET had a sensitivity of 0.86 (95% CI, 0.88–0.93) and a specificity of 0.88 (95% CI, 0.84–0.91) in the restaging of RCC, especially in the diagnosis of distant metastases [23].

Moreover, high baseline [^18^F]FDG uptake in advanced renal cell carcinoma correlates with disease aggressiveness and poor prognosis [24,25].

FDG-PET is also a useful “imaging biomarker” for patients with advanced RCC planning sequential molecular targeted therapies [26].

However, the value of FDG-PET in the initial staging of RCC is debated, mainly because of masking due to urinary tracer excretion [27,28], even if tracer uptake has been recently demonstrated in most patients with RCC [25].

During recent years, interest has grown in non-FDG radiopharmaceuticals for PET imaging that target different processes other than glucose metabolism and that add tumor typing information useful for individualized treatments.

The expression of the prostate-specific membrane antigen (PSMA) on the neo-vasculature of many solid tumors including RCC [29,30] has gained attention as a potential target for PSMA-based imaging. Importantly, adjacent normal endothelium does not express PSMA [29,31], suggesting that PSMA expression by neovessels may be induced by tumor-related factors.

PSMA is a 750 amino acid, type II transmembrane glycoprotein (100–120 kDa) with an intracellular component, a transmembrane component, and a large extracellular domain, that has been shown to be highly expressed in the proximal tubules of normal kidney tissues and is differentially expressed in the tumor-associated neo-vasculature of different renal tumors [30].

In particular, 76.2% of ccRCC and 31.2% of chromophobe RCC samples show expression of PSMA glycoprotein, whereas papillary RCC samples are negative [30]. Moreover, PSMA hyper-expression is correlated to overall survival among patients with ccRCC [32].

Due to the enzymatic activity of PSMA, it was possible to develop specific inhibitors targeting the extracellular domain of glycoprotein [33], from which a number of different PSMA-targeted PET tracers have been derived [33,34].

The most widely used PSMA-ligand for PET-imaging in Europe is a ^68^Ga-labelled PSMA inhibitor Glu-NH-CO-NH-Lys(Ahx)-HBED-CC ([^68^Ga]-PSMA-HBED-CC) [35].

The specific Monograph “Gallium (^68^Ga) PSMA-11 injection” has been recently published in the European Pharmacopeia [36] and the FDA approved ^68^Ga-PSMA-11 injection in December 2020 [37].

PSMA-based tracers have been mainly assessed for their diagnostic performance, often in retrospective studies [38,39,40,41] or in small sample-size prospective studies [42,43,44]. Quantitative assessment of PSMA-based PET signals expressed as SUV metrics has been correlated with pathology characteristics of RCC [43], but only a few studies analyzed the correlation between tracer uptake and PSMA expression at IHC analysis [45,46,47]. However, these studies report results of a small subgroup of samples [45,46] or use the PET signal to distinguish between benign and malignant renal mass [47]. Furthermore, a correlation between SUV values and the grade of PSMA staining results is controversial [45,46]. There is a growing interest in focusing on the impact of PSMA-based PET/CT in patient management to identify oligometastatic cancer [46,48,49] or in preoperative staging [43]. However, all the aspects that can be derived from PSMA-PET/CT are not evaluated altogether.

The aim of this study was to describe PSMA-PET/CT results in a retrospective cohort of patients with RCC submitted to the PET scan after non-conclusive results of conventional imaging. Moreover, we assessed and scored the tissue expression of PSMA in surgical or biopsy tumor specimens and correlated it with the intensity of the PET signal. Finally, we evaluated the predictive role of PSMA-PET/CT and its impact on patient management.

## 2. Methods

From January 2018 to December 2022, 295 patients diagnosed with RCC were surgically treated in our institution and managed by the uro-oncologic multidisciplinary board. Of them, 31 subjects were submitted to PSMA PET/CT for staging or restaging due to non-conclusive results of conventional imaging (contrast-enhanced CT, ultrasonography, magnetic resonance). In the same period, 2.503 PSMA PET/CT scans were performed in our PET center. The PSMA expression in tumor tissue was tested on specimens obtained by surgery or metastasectomy using anti-PSMA immunostains in 26 patients. Only cases with immunohistochemistry evaluation of PSMA expression were included in the study.

Results of PSMA-PET/CT were confirmed by MR, CT, biopsy, or follow-up.

The study was approved by the Ethics Committee of our Institution (reference number: N°1051/2018/TESS/AOUPR) and was conducted according to the principles of the Declaration of Helsinki and the principles of Good Clinical Practice.

### 2.1. Imaging Protocol

A whole body PET/CT was acquired from the vertex to the medium thigh of the femur 60 min after I.V. injection of [^68^Ga]-PSMA-HBED-CC (150 MBq) on a hybrid scanner Discovery IQ (GE Healthcare^®^, Chicago, IL, USA). A low-dose non-enhanced CT was performed for attenuation correction and anatomical correlation. All PET images were corrected for attenuation, dead time, random events, and scatter. The reconstruction of PET images was performed with an iterative algorithm (ordered-subset expectation maximization). Synthesis of [^68^Ga]-PSMA-HBED-CC was performed using a fully automated module (Scintomics GRP^®^, Fuerstenfeldbruck, Germany) and a pharmaceutical grade ^68^Ge/^68^Ga generator (1850 MBq, GalliaPharm^®^ Eckert & Ziegler, Berlin, Germany) as previously described [50]. The mean radiochemical purity of [^68^Ga]PSMA-HBED-CC was 99.90% and the mean yield of the labeling reaction was 68.71%.

Two experienced (>500 PSMA PET/CT a year) board-certified nuclear medicine physicians (MS, GB) evaluated all the images visually and semiquantitatively. Focal tracer uptake was considered suspicious for cancer lesions when higher than in the reference region. Background activity was assessed in the gluteus maximus muscle which was considered as the reference region. A PET scan was assessed as negative when abnormal sites of tracer uptake were undetectable.

Semiquantitative analysis of tracer uptake was performed using spherical volumes of interest (VOIs) which were semi-automatically drawn on orthogonal planes. The Maximum Standard Uptake Value (SUVmax) was measured in the target nodal or distant lesion which was considered as the hottest lesion.

Tumor to background ratio (TBR) was measured as the ratio between SUV in the hottest lesion and SUV in the reference region.

### 2.2. Pathological Assessment of PSMA Expression

Tissue specimens were obtained by surgery (26 nefrectomy; 6 metastasectomy). PSMA expression in tumor tissues was tested using anti-PSMA immunostains (1:100, EPI92 clone, Epitomics). In selected cases, immunostains were also performed to confirm the histological subtype (anti-carbonic anhydrase IX, AMACR, CD10, and cytocheratin 7).

The intensity of PSMA expression was scored into four categories: 0: none; 1: weak; 2: moderate; 3: strong. PSMA expression was also graded according to the proportion of vessels involved (PSMA%) into four categories: 0: none; 1: 1–25%; 2: 25–50%; 3: >50%). Intensity and percentage expression were added to yield a combined score (0–6). Samples with combined scores 0–2 were considered absent or mildly positive, 3–4 moderately positive and 5–6 strongly positive. PSMA scores were used for correlation with PET scan results.

### 2.3. Statistical Analysis

Descriptive statistics were used to display patient data as mean and median, as well as percentages. Quantitative data (age, SUVmax, TBR, IHC-PSMA scores) were synthesized by calculating the mean (with range interval) and median values. Categorical data (PET positivity, classification and pathologic TNM staging, PSMA expression) were shown as frequencies and percentages. The *p*-values test was used to assess the group differences significance with a two-tailed ANOVA. The level of significance was set at *p* < 0.05. Survival curves were estimated using the Kaplan-Meier method and were compared using the long-rank test. OS was measured from the date of the PET exam to the date of death from any cause. Interrater reliability for PSMA PET/CT was measured as percent agreement and Cohen’s kappa. For all the analyses the statistical software used was GraphPad Prism (GraphPad Prism 5 for macOS).

## 3. Results

We evaluated the PSMA PET/CT scans of 26 patients with RCC submitted to the exam during disease staging (4) or restaging (22): 10 females, 16 males, mean age 65.6 years (range 42–84). Their main clinical characteristics are reported in Table 1.

The tumor characteristics at initial diagnosis are reported in Appendix A.

IHC assessment of PSMA expression was available for all of them.

The interval between PET/CT and surgery of the primary tumor or metastases ranged from 1 to 1369 months (median value 69 months).

### 3.1. Immunohistochemistry Patterns

Immunostains (IHC) for PSMA were performed in 22 ccRCC, 2 papillary RCC, 1 cromophobe, and 1 “not otherwise specified” RCC).

In all patients, IHC was performed on the primary tumor and in 6 patients also on the paired metastasis (2 lung metastases, 1 lymph node, 1 cerebellum, 1 adrenal, 1 liver metastases).

Sixteen tumors were strongly PSMA-positive on tumor vessels (as expressed by a combined score of percentage and intensity of PSMA reactivity), 5 were moderately positive and 5 had absent/mild positivity. Twenty of 22 cc-RCC scored moderately (5) or strongly (15). The two papillary and the single NAS RCC were in the lower category score; the chromophobe RCC scored strongly (*p* = 0.016, clear cell vs. non-clear cell RCC).

In all six cases examined on both the primary tumor and the metastasis, the combined PSMA score was concordant. The interval between the primary resection and the metastases ranged from 20 to 99 months.

### 3.2. PSMA-PET Signal

PET scan was positive in 17 patients (65%) and negative in 9 patients (35%).

The primary tumor was detected in all patients submitted to the exam for staging (mean SUVmax 19.3, range (5.45–54.2).

Distant metastases were detected in the skeleton (4 patients), lungs (9 patients), liver (2 patients), inferior vena cava (2 patients), brain (2 patients), adrenal gland (2 patients), soft tissue (2 patients) and thyroid (1 patient). One example is reported in Figure 1.

Lymph node metastases were detected in 5 patients. Unknown metastatic sites were identified in two patients in the cerebellum and frontal brain cortex (Figure 2 and Figure 3) as well as unknown vascular infiltration of the inferior vena cava in two patients.

In the case of bone involvement, extension and number of lesions were greater than in the CT scan (median number of bone lesions on PET/CT 3.5).

Results of PSMA-PET/CT were confirmed with MR, CT, biopsy, or follow-up.

Quantitative analysis of tracer uptake in the target lesion (Appendix A) showed an average SUVmax value of 30.7 and a median value of 24.03 (range 4.4–114.6). The lowest SUVmax values (<10) were registered in seven patients with a median value of 7.4.

The mean value of TBR was 36.6, with an average SUVmax in the reference region (gluteus maximus muscle) of 0.92 [range 1.7–0.6].

SUVmax and TBR values showed the same behavior in positive scans (Appendix A).

The most intense lesion was detected in recurrence tissue in the renal fossa (SUVmax 114.6). The most intense distant metastases were revealed in the liver and bone (48.6 and 46.3 respectively).

Reporter agreement was very high for both distant metastasis location and local recurrence (kappa 1, 100%).

### 3.3. PET Results and Cancer Type

In the subgroup of cc-RCCs (*n* = 22) PSMA-PET/CT was positive in 15 patients (68%): mean and median values of SUVmax in the target lesion were 34.1 and 24.9 [range 4.4–114.6] respectively, and the mean was TBR 41.7 [range 5.6–143].

Metastatic lesions were detected in 14 of 22 cc-RCCs (mean number of lesions per patient 3.6).

Mean values of SUVmax across different sites of metastases are reported in Table 2, with the highest SUVmax values detected in bone (28.61) and liver (29.81) lesions.

PSMA-PET/CT resulted negative in one case of papillary cancer type II while showing moderate uptake in the second case, in the target lesion (iliac bone, SUVmax 10.2, TBR 6) and metastatic sites (SUVmax 4.5 in the pulmonary lesion and SUVmax 7.7 in the abdominal lymph nodes. Values of SUVmax in the target lesion resulted lower than in ccRCC similar sites (Figure 4).

In the patient with chromophobe cancer, a weak PET signal was detected (SUVmax 5.45, TBR 5.62) in the target lesion (left kidney) without evidence of distant metastases.

### 3.4. PSMA Expression on IHC and PET Results

All positive PSMA-PET/CT results (Figure 5a) were revealed in strongly or moderate PSMA-positive cases at immunohistochemistry (combined scores 3–4 and 5–6).

The SUVmax value was found to be significantly correlated with the intensity of PSMA expression (*rs*: 0.33; *p* = 0.01), while no correlation was found between SUVmax and PSMA% (*rs*: 9.097 × 10^−6^; *p* = 0.99) neither with the SMA combined score (*rs*: 0.07; *p* = 0.3).

Distant metastases were detected using PSMA-PET/CT in tumors with PSMA scores 3–6.

In cc-RCC cases, most positive PET scans (12 of 15) scored 5–6 at IHC and the remaining scored 3 (Figure 5b).

The patient with papillary cancer and a positive PSMA-PET/CT presented a grade 3 combined PSMA score as well as the chromophobe cancer with positive PSMA-PET/CT which scored strongly (combined PSMA score 5).

The intensity of PSMA expression was elevated (grade 2–3) in 16/17 patients with positive PSMA-PET/CT (Figure 6a) (14 ccRCC, 1 papillary RCC, 1 chromophobe RCC). Only one case of ccRCC with a positive PET had a grade 1 intensity but a grade 2 PSMA%.

The proportion of vessels involved (PSMA%) showed elevated results (grade 2–3) in 14/17 patients with a positive PSMA-PET/CT (Figure 6b): 13 ccRCC and 1 chromofobe RCC. Both papillary cancers presented a grade 1 PSMA%.

### 3.5. Impact of PSMA-Directed PET/CT on Management

Results of PSMA-PET/CT changed treatment planning in several ways (Table 3) also taking into account the metastatic site.

The most relevant change was the execution of stereotactic ablative radiotherapy (SABR) or surgery in patients with brain lesions (*n* = 2) and the most frequent change (*n* = 11) was the cancellation of surgical or ablative intervention on a specific metastatic lesion due to detection of multiple metastatic sites. Metastasis-directed surgery was planned in two patients with a single metastatic lesion (adrenal gland and lung) (*n* = 2). Systemic therapy was postponed in case of diffuse metastases (*n* = 1).

### 3.6. Survival Analysis

After a median follow-up of 33 months (range 1–69) following the PET scan, 16 patients were dead and 10 alive. In 15 of 16 cases, death was related to the neoplastic disease. The survival curve of the study population is reported in Figure 7a.

The study results showed that the median OS was 33 months in the ccRCC group and 44 months in the other tumor types (HR 1.2, 95% CI 0.30–4.81, *p* = 0.79) (Figure 7b).

In patients with negative PSMA-PET/CT, the median survival was 48 months, and 24 months in patients with a positive scan (HR 3.6, 95% CI 1.36–9.79, *p* = 0.001) (Figure 8).

Patients with SUVmax of the target lesion ≥ 7.4 showed lower survival (24 months) compared to patients with SUVmax < 7.4 (33 months) (HR 0.28, 95% CI 0.10–0.77, *p* = 0.01) (Figure 9).

In the subclassification according to the combined score (Figure 10a), median OS was 45.5, 18, and 34 months in low (0–2), moderate (3–4), and strong (5–6) categories, respectively (*p* = 0.1). Grouping patients with moderate-strong combined score (3–6) median survival resulted in 45.5 months in the low score group 0–2 and 33 months in the moderate-strong score group (HR 1.6, 95% CI 0.47–5.74, *p* = 0.48) (Figure 10b).

In ccRCC patients (Figure 11a,b) median survival was 37.5, 24, and 34 months in the 0–2, 3–4, and 5–6 score categories, respectively (HR, 95% CI *p* = 0.2). By SUVmax value ≥ 7.4, median survival was 24 months, and 51.5 months for SUVmax < 7.4 (HR 0.29, 95% CI 0.10–0.83, *p* = 0.03).

## 4. Discussion

In our study, all positive PSMA-PET/CT results were revealed in patients with strong or moderate PSMA expression at immunohistochemistry (Figure 5) confirming the strict relationship between target expression in tissue and imaging signal as previously described [45].

SUVmax values correlated to the intensity of PSMA expression which were assessed using IHC (*p* = 0.01), especially in the ccRCC subgroup (*p* = 0.009), while no significant correlation resulted with the proportion of vessels involved, or the combined score (intensity added to the percentage expression). This finding may suggest that the PET signal could depend on a binary “on-off” mechanism by the presence or absence of the imaging target in the tumor tissue and that the SUV value could be consistent with PSMA-dependent signaling rather than the extent of the neovasculature. Gühne et al. [45] didn’t find any correlation between SUV values and PSMA expression, but they included in their study a smaller number of patients.

In this study clear cell renal cell carcinoma showed the most intense and extensive expression of PSMA (Figure 5), followed by chromophobe renal cell carcinoma (combined score 5), whereas papillary renal cell carcinoma showed the least in both extent and intensity of PSMA expression (combined score 0–2), as shown in previous studies [32,40,51].

The PET result was strongly correlated with survival, confirming the potential of PSMA-targeted PET as an aggressiveness biomarker (Figure 8a), while the expression of PSMA in tissue specimens could not discriminate towards aggressive phenotypes of RCC (Figure 10a,b).

Using a specific cut-off value of SUVmax ≥ 7.4, PSMA-PET/CT demonstrated a strong prognostic potential for discriminating patients with a poor prognosis (*p* < 0.001, Figure 9).

A similar subclassification using the SUVmax threshold of patients with RCC was performed in a previous paper by Nakaigawa et al. [26] to assess the predictive role of FDG-PET after the first molecular targeted therapy.

Our study has also confirmed that papillary RCC, and especially type II, is not avid of ^68^Ga-PSMA as recently demonstrated in a large multicenter cohort by Zschäbitz et al. [52] probably for decreased neovascularity in comparison with ccRCC.

Indeed, Bauman and colleagues [53] showed that within RCC morphotypes, the CD105/CD31 ratio, used to measure neovascularity, was significantly lower in papillary tumors (*p* = 0.05) when compared to clear cell samples.

However, papillary RCC lesions may be detected and monitored by FDG-PET [54].

Although it is not recommended for the management of RCC patients, FDG-PET is often used in the restaging setting, mainly because of the high rates of recurrence combined with the necessity of early detection to increase the chance of favorable outcomes, especially in confined lesions.

Some recent reports have shown that PSMA-PET may potentially be more sensitive than FDG-PET in metastatic RCC, emerging as a new imaging tool in patients with oligometastatic disease receiving definitive local therapies [40,54,55].

Such novel imaging approaches that facilitate the detection of changes in tumor biology may form the basis for improved predictive biomarkers and tailored therapy.

In the paper of Morgantetti et al. [56] PSMA expression resulted higher in ccRCC vena cava tumor extensions compared with the renal tumor mass. Therefore, Authors speculate that PSMA activity may be a useful marker for neoangiogenesis providing information on the invasive nature of RCC.

The need for a new assessment of RCC focusing not only on the tumor volume but also on adding phenotypic information for patient/tumor-specific treatment strategies has emerged [57].

New therapies are beginning to demonstrate promising results in the context of metastatic RCC and imaging biomarkers play a central role in monitoring treatment response.

Therefore, it is crucial to understand the presentation of patients with RCC and to correctly assess disease burden in order to correctly plan treatment, particularly to distinguish patients with advanced disease and poor prognosis, from ones within the oligometastatic setting candidate to curative treatment.

It has been recently demonstrated that PSMA PET/CT led to a change in management in 49% of the patients with metastatic renal cancer and detected additional metastases compared to CT in 25% of the cases [48].

A recent retrospective study has shown a higher impact of PSMA-PET/CT on disease management for patients with recurrent than de novo metastatic disease (31% vs. 23%, *p* = 0.73 [49]. However, the study does not report a correlation with a grading score of PSMA expression in tissue specimens.

In our study, the most frequent change in patient management induced by PSMA PET/CT was the cancellation of surgical or ablative intervention on metastatic lesions due to the detection of diffuse disease, and the most relevant was the execution of stereotactic ablative radiotherapy or surgery in patients with a brain lesion (Table 3).

As a final point, RCC represents a challenging disease setting for imaging experts due to its ability to metastasize in a number of unusual sites: agreement in assessing images is therefore crucial to guarantee repeatability. In this study a very high inter-rater reliability in reading PSMA-PET/CT scans has been measured for both distant metastasis location and local recurrence (kappa 1, 100%), guaranteeing imaging accuracy.

One limitation of this study is the assessment of PSMA expression at IHC in different samples, as the primary tumor or the metastatic lesions, which may be quite different in PSMA expression. However, in all six cases examined on both the primary tumor and the metastasis, the combined PSMA score was concordant.

Then, the retrospective nature of the analysis and the execution of the exams in cases of equivocal or non-conclusive conventional imaging make the population non-homogeneous and may have biased the results. However, the subjects included in the study were evaluated consecutively using both PSMA-PET/CT and IHC.

Finally, the small sample size can make it difficult to identify significant relationships in the data and the obtained results require confirmation in prospective and larger studies. Taking into account these considerations, in this real-world series PSMA-PET/CT has shown a high prognostic value and impact on the treatment management of patients with RCC.

## 5. Conclusions

In patients with renal cell cancer PSMA-PET/CT signal is strongly correlated with survival confirming its potential as an aggressiveness biomarker. PSMA-PET/CT signal is strictly related to the intensity of PSMA expression in tissue, especially in ccRCCs.

Unknown sites of metastatic disease are detected by PSMA-PET and bone involvement revealed by PET is greater than in conventional imaging, with relevant changes in patient management.

## Figures and Tables

**Figure 1 diagnostics-13-03082-f001:**
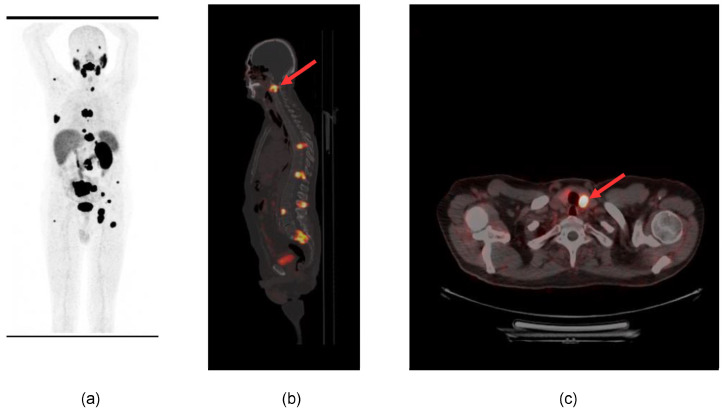
^68^Ga-PSMA PET/CT scan in restaging ccRCC. (**a**) MIP; (**b**) multiple bone lesions in the spine: the arrow shows the lesion in the second cervical vertebra (sagittal section of fused PET/CT); (**c**) single lesion in the left lobe of the thyroid gland (the arrow shows focal increased tracer uptake in the thyroid nodule) (axial PET/CT section).

**Figure 2 diagnostics-13-03082-f002:**
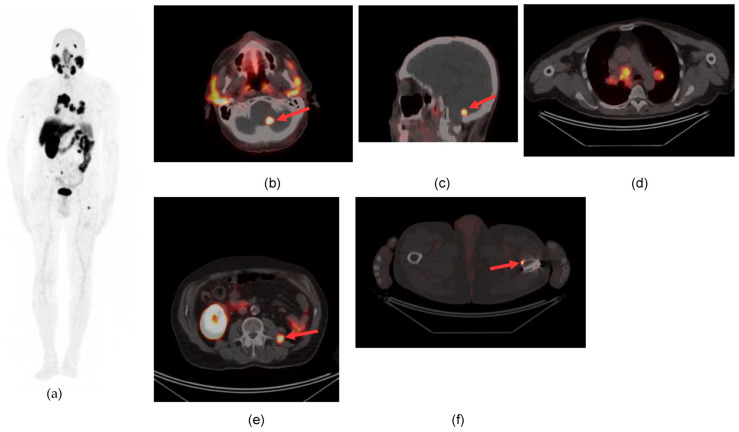
^68^GaPSMA PET/CT detects an unknown cerebellar metastatic lesion (arrow) in a patient with metastatic ccRCC. (**a**) MIP; (**b**,**c**) focal tracer uptake in the left cerebellar hemisphere (axial and sagittal sections of PET/CT) (**d**) carinal and hilar nodal involvement; (**e**) solid lesion in the abdomen (arrow) near the left iliopsoas muscle; (**f**) single bone lesion (arrow) in the cortical profile of the femoral diaphysis ((**d**–**f**) axial sections of PET/CT).

**Figure 3 diagnostics-13-03082-f003:**
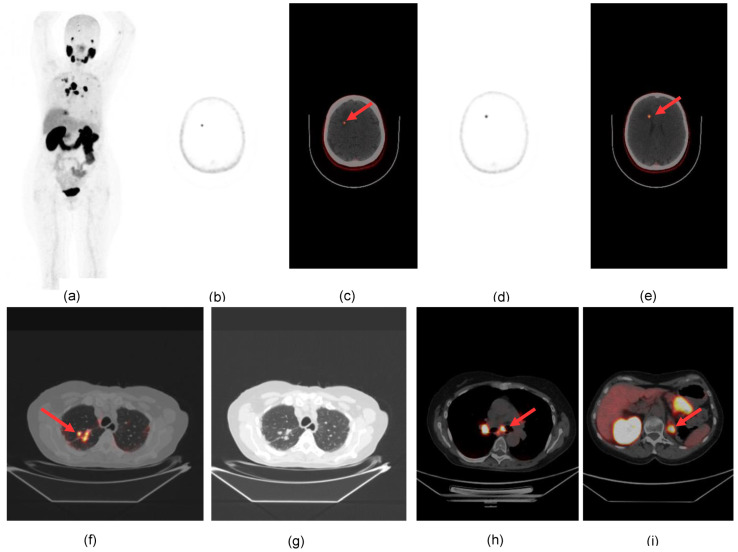
^68^Ga-PSMA PET/CT reveals unknown brain lesions in ccRCC: (**a**) MIP; (**b**–**e**) axial sections of emission PET (**b**,**d**) and fused PET/CT (**c**,**e**) showing two brain lesions in the right frontal cortex (arrows); (**f**,**g**) fused PET/CT and CT axial sections of the thorax showing parenchimal lesions in the upper lobe of the right lung; (**h**) carinal (arrow) and hilar nodal involvement; (**i**) metastatic lesion of the left adrenal gland (arrow).

**Figure 4 diagnostics-13-03082-f004:**
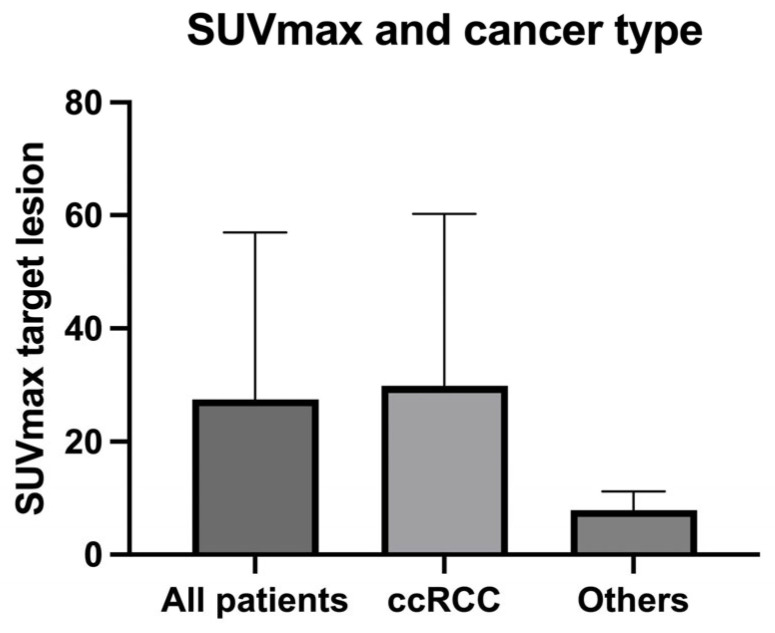
SUVmax in the target lesion according to cancer type in the enrolled patients.

**Figure 5 diagnostics-13-03082-f005:**
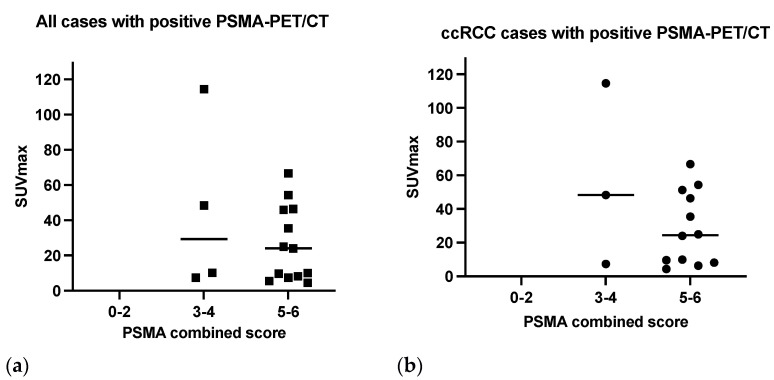
Combined PSMA score at immunohistochemistry in patients with positive PSMA-PET/CT: (**a**) all PET-positive cases; (**b**) ccRCC cases with positive PSMA-PET/CT.

**Figure 6 diagnostics-13-03082-f006:**
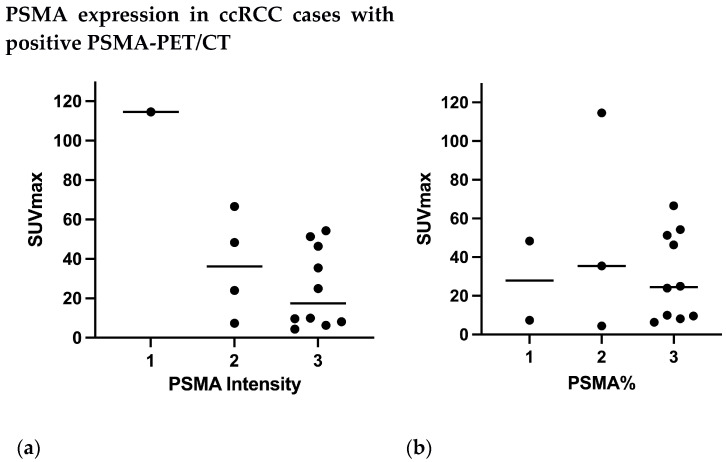
(**a**) Intensity score of PSMA expression in ccRCC cases with positive PET (**b**) Proportion of vessels involved (PSMA%) in ccRCC cases with positive PET.

**Figure 7 diagnostics-13-03082-f007:**
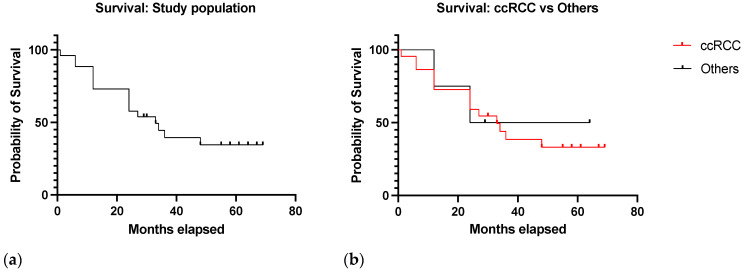
(**a**) Survival curve in the study population (**b**) Survival curves in the ccRCC group compared to other tumor types.

**Figure 8 diagnostics-13-03082-f008:**
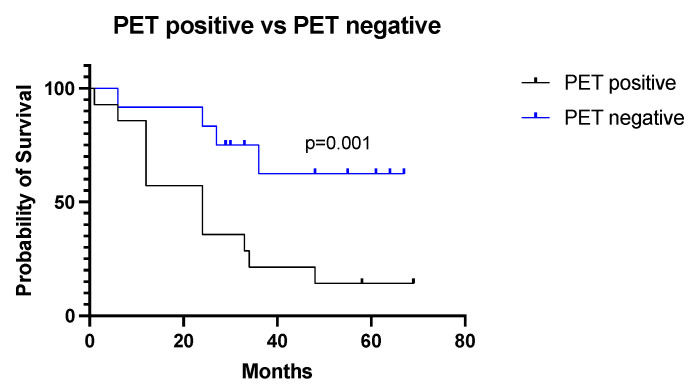
Survival curves according to PSMA-PET/CT results in the study population.

**Figure 9 diagnostics-13-03082-f009:**
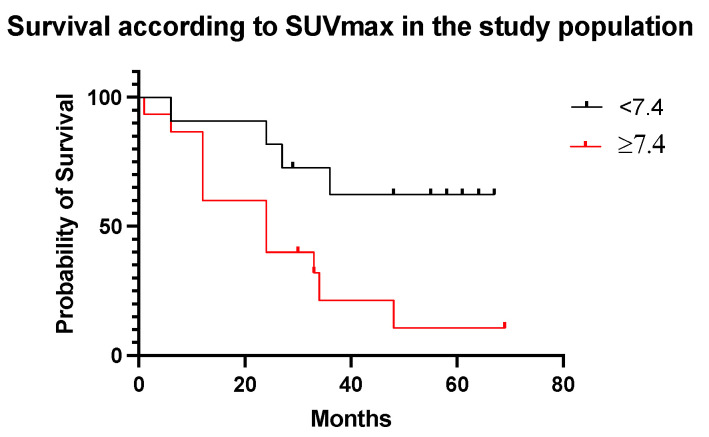
Survival curves according to SUVmax cut-off value in the study population.

**Figure 10 diagnostics-13-03082-f010:**
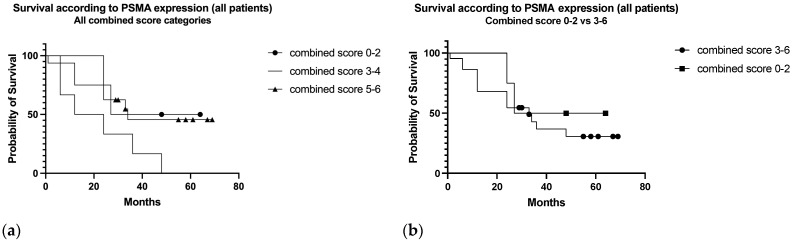
Survival curves according to the PSMA combined score categories: (**a**) all the score categories; (**b**) grouped categories comparing moderate-strong (3–6) and low (0–2) combined PSMA score categories (*p* = 0.48).

**Figure 11 diagnostics-13-03082-f011:**
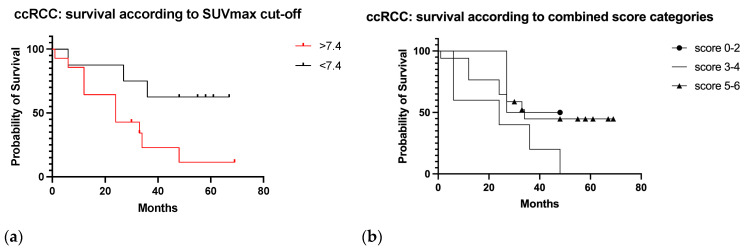
Survival curves in the ccRCC cases according to SUVmax cut-off value in the target lesion (**a**) and according to the combined score categories (**b**).

**Table 1 diagnostics-13-03082-t001:** Clinical characteristics of the study patients with renal cell cancer included in the study.

Patient’s Characteristics	Data
Mean age (range, yrs)	65.6 (42–85)
No. male (%)	16 (61.5)
No. female (%)	10 (38.5)
Median months follow up (range)	29.5 (1–69)
No. follow up end status (%)	
Alive (No.)	10 (38.5)
Dead (No.)	16 (61.5)
No. histological subtype (%)	
Clear cell	22 (84.6)
Papillary	2 (7.7)
Chromophobe	1 (3.8)
NAS	1 (3.8)
No. primary intervention (%)	
Radical nephrectomy	22 (84.6)
Partial nephrectomy	2 (7.69)
RFA	1 (3.85)
None	1 (3.85)
No. patients on therapy	
Current	4
Before PSMA-PET/CT scan	7
None	15

**Table 2 diagnostics-13-03082-t002:** Site and SUVmax of metastatic lesions with tracer uptake on PSMA-PET/CT.

Site of Metastatic Lesions at PSMA PET/CT in cc-RCC	No. Patients	SUVmax Mean Value (Range)
Lung	7	12.08 (3.2–24.9)
Liver	2	29.8 (11.2–48.3)
Bone	3	24.0 (10.0–46.3)
Lymph nodes	4	20.6 (7.6–35.4)
Brain	2	15.5 (10.7–20.2)
Adrenal gland	2	12.8 (9.6–15.9)
Vascular	2	11.8 (5.8–17.8)
Soft tissue	2	36.1 (5.6–66.6)
Retroperitoneum	2	22.8 (26.7–18.9)
Thyroid	1	24.5

**Table 3 diagnostics-13-03082-t003:** Impact of PSMA-PET/CT results on treatment planning.

No. Patients	Planned Therapy	Performed Therapy
4	systemic	SABR or surgery +/− systemic
11	surveillance or local treatment (surgery, RT, radiofrequency ablation)	systemic
4	surveillance	surgery or systemic
1	systemic	palliation

## Data Availability

Raw data that support the findings of this study are not publicly available to preserve the privacy of research participants. However, they are available on request, from the corresponding author [S.M.].

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
