# Peer review of "High Prognostic Value of 68Ga-PSMA PET/CT in Renal Cell Carcinoma and Association with PSMA Expression Assessed by Immunohistochemistry"

_diagnostics, 2023, doi:10.3390/diagnostics13193082_

Round 1

Reviewer 1 Report

Thank you for your interesting submission.

The following suggestions could be considered:

Title: Consider omission of “superior” from the article title.

With regards to the methodology, the following:

Clear in-and exclusion criteria needed.

How were potential false positive findings on PSMA PET/CT dealt with (eg lymph nodes, thyroid lesion, lung lesions)?

Timing of scans in relation to treatment received seems problematic (1-1369 months!)

Limitations: only those patients with inconclusive findings on conventional imaging were referred for PSMA PET, which may have biased the results.

Other comments:

Replace “Before PSMA PET/CT scan” in the table with “Baseline PSMA PET/CT”

Correct “systemic” to “systemic”

Consider including some of the more recent relevant publications as references:

Pozzessere C, Bassanelli M, Ceribelli A, Rasul S, Li S, Prior JO, Cicone F. Renal cell carcinoma: the oncologist asks, can PSMA PET/CT answer?. Current urology reports. 2019 Nov;20:1-0.

Muselaers S, Erdem S, Bertolo R, Ingels A, Kara Ö, Pavan N, Roussel E, Pecoraro A, Marchioni M, Carbonara U, Marandino L. PSMA PET/CT in renal cell carcinoma: an overview of current literature. Journal of Clinical Medicine. 2022 Mar 25;11(7):1829.

Urso L, Castello A, Rocca GC, Lancia F, Panareo S, Cittanti C, Uccelli L, Florimonte L, Castellani M, Ippolito C, Frassoldati A. Role of PSMA-ligands imaging in Renal Cell Carcinoma management: Current status and future perspectives. Journal of Cancer Research and Clinical Oncology. 2022 Jun;148(6):1299-311.

Oflas M, Ozluk Y, Sanli O, Ozkan ZG, Kuyumcu S. 68Ga-PSMA uptake patterns of clear cell renal carcinoma across different histopathological subtypes. Clinical nuclear medicine. 2022 Jan 1;47(1):e45-6.

Sasikumar A, Joy A, Nanabala R, Unni M, Padmanabhan TK. Complimentary pattern of uptake in 18F-FDG PET/CT and 68Ga–prostate-specific membrane antigen PET/CT in a case of metastatic clear cell renal carcinoma. Clinical Nuclear Medicine. 2016 Dec 1;41(12):e517-9.

Ahn T, Roberts MJ, Abduljabar A, Joshi A, Perera M, Rhee H, Wood S, Vela I. A review of prostate-specific membrane antigen (PSMA) positron emission tomography (PET) in renal cell carcinoma (RCC). Molecular imaging and biology. 2019 Oct;21:799-807.

Pozzessere C, Bassanelli M, Ceribelli A, Rasul S, Li S, Prior JO, Cicone F. Renal cell carcinoma: the oncologist asks, can PSMA PET/CT answer?. Current urology reports. 2019 Nov;20:1-0.

Raveenthiran S, Esler R, Yaxley J, Kyle S. The use of 68 Ga-PET/CT PSMA in the staging of primary and suspected recurrent renal cell carcinoma. European Journal of Nuclear Medicine and Molecular Imaging. 2019 Oct 1;46:2280-8.

Minor spelling and grammatical issues

Author Response

Please see the attachment, point-by-point response

Reviewer 2 Report

PSMA-PET signal predicts a poor prognosis confirming its potential as aggressiveness biomarker and provides paramount additional informations influencing patient management. Several comments listed below.

1.      Line 47, It is unclear whether the author's something new in this work. According to the evaluation, several published literature by other researchers in the past adequately explain the issues you made in the present paper. Please be careful to highlight in the introduction section anything really innovative in this work.

2.      Line 336, please give comprehensive discussion of PSMA combined with PSMA-PET in positive.

3.      Line 354, for making easily to understanding in a and b for intensity score and proportion should be put in below. Not in the upside.

4.      Line 344, why figure 6 not combined with Figure 5 since it is similar?

5.      Line 350, the authors mention 14 ccRCC, 1 papillary RCC, 1 chromofobe RCC, please recheck again.

6.      Line 364, foe giving the caption of Impact of PSMA PET/CT results on treatment planning should put below the table.

-

Author Response

(The authors gave the same response as above.)
